# Executive dysfunction is associated with altered hippocampal-prefrontal functional connectivity in 3xTg Alzheimer's model mice

Grace Cunliffe [1,2], Li Yang Tan[2,3], Sangyong Jung[2,3], Jonathan Turner[1] & John Gigg [1] ✉

Executive dysfunction encompasses altered decision-making, attentional deficits, excessive risk-taking behaviours, and inefficient planning. Alongside memory loss, these cognitive deficits are amongst the most frequently reported symptoms of Alzheimer's disease (AD). Normal executive function depends on connectivity between the ventral hippocampus and medial prefrontal cortex (mPFC), yet abnormalities in this circuit and how such changes lead to cognitive dysfunction in AD have yet to be fully elucidated. Here, we show that 6-month-old male 3xTg AD mice display maladaptive decision-making in the rodent 4-Choice Gambling Task measure of executive function. Extracellular field recordings in the infralimbic cortex of age-matched 3xTg mice show layer-specific reductions in response amplitude and paired-pulse ratio compared to controls following activation of hippocampal input fibres, indicating changes to short-term hippocampal-prefrontal synaptic plasticity. These results therefore reveal a pre-clinical deficit in executive function that correlates with mPFC synaptic plasticity deficits in a mouse model for AD.

Ventral hippocampal (vHIP) inputs to the medial prefrontal cortex (mPFC) are essential for controlling executive components of cognition, including attention, decision-making, risk-taking behaviours, and behavioural flexibility[1–7]. These inputs are glutamatergic, arising in the CA1 and subiculum regions of the vHIP and terminating predominantly in layers II/III and V of the infralimbic (IL) and prelimbic (PL) cortices of the mPFC[8–11]. Both the prefrontal cortex and hippocampus display substantial levels of neurodegeneration and amyloid beta (Aβ) plaque and tau neurofibrillary tangle (NFT) accumulation over the course of Alzheimer's disease (AD), so it is unsurprising that alongside declarative memory loss, cognitive deficits underlying executive dysfunction are among the most frequently reported symptoms associated with disease progression[3,12–15]. AD is the most common form of dementia, currently affecting over 55 million people worldwide, and is estimated to be prevalent in 5–8% of over 60s[16–18]. Nevertheless, no cure exists and current treatment options act simply to slow the spread of disease. Studies have shown that hippocampal-prefrontal connectivity is weakened in AD rodent models[19], but changes in functional connectivity between the vHIP and mPFC and how these may lead to executive dysfunction over the course of the disease have yet to be fully elucidated.

The examination of executive function in an AD context requires a test that incorporates aspects of decision-making, such as behavioural flexibility, risk-taking tendencies, and planning for long-term gain, whilst minimising handling and related stress to the animal. Accordingly, the rodent touchscreen-operant 4-Choice Gambling Task (4CGT) was chosen for this study. This task is based on the clinically validated Iowa Gambling Task (IGT), which measures executive function in human patients. Patients with damage to the dorsolateral prefrontal cortex (dlPFC), analogous to the mPFC of rodents, fail to achieve optimal decision-making in the IGT[20,21], as do AD patients[22–24]. Both the 4CGT and IGT employ the concept of optimal choice preference for low-risk, low penalty options that maximise reward over time, versus high-risk, high-reward choice preference, which overall leads to longer punishment duration and reduced reward availability[25]. The 4CGT employs touchscreen trials in which four lit square stimuli are presented, each associated with variable volumes of reinforcement (milkshake delivery) and punishment duration (luminance inversion). This test has been deployed effectively many times in rat models[25–28] and a small number of studies have shown that mice are also capable of learning and performing the task[29–32]. To our knowledge, no preclinical studies have applied the

[1]Division of Neuroscience, School of Biological Sciences, Faculty of Biology, Medicine and Health, The University of Manchester, Manchester, UK. [2]Institute of Molecular and Cell Biology (IMCB), Agency for Science, Technology and Research (A*STAR), Singapore, Singapore. [3]Department of Medical Science, College of Medicine, CHA University, Seongnam, Republic of Korea. ✉e-mail: J.Gigg@manchester.ac.uk

4CGT in an AD mouse model. The triple transgenic (3xTg) mouse[33] is a widely used model for AD as it recapitulates both Aβ plaque and tau NFT pathology; the two major hallmarks of the disease. Previous studies have shown that these pathological hallmarks develop relatively slowly in 3xTg mice, only becoming overtly visible from 12 months of age. As synaptic alterations and cognitive deficits occur as early as 4 months[33,34] due to the presence of intraneuronal amyloid, an advantage of this model is that it provides a lengthy prodromal period over which brain and behaviour changes can be measured that relate to early AD in patients, that is, prior to the accumulation of overt pathological proteins. This period represents the most likely target for disease-modifying interventions.

Studies in AD mouse models and AD patients have reported modifications to the synaptic excitatory/inhibitory (E-I) balance in the prefrontal cortex due to alterations in glutamatergic and GABAergic receptor expression, metabolism, and transport[35–40]. These observations occur early in disease progression, resulting in instability of the neural network, which may potentiate disease pathogenesis and resultant cognitive disturbances[41–43]. The paired pulse ratio (PPR) is commonly used to assess short-term synaptic plasticity dependent on the probability of presynaptic vesicular transmitter release. This process is governed by presynaptic calcium dynamics and vesicle release machinery underlying vesicle exocytosis, requiring the recruitment of numerous membrane-associated proteins including synaptotagmins, complexins, and SNARE proteins[44]. Alterations to calcium dynamics and the expression of proteins associated with vesicle exocytosis are reported to contribute towards synaptic deficits in the hippocampus and frontal regions of AD patients[45–51]. However, how AD pathology impacts short-term synaptic plasticity and local field potential (LFP) generation in the vHIP-mPFC pathway specifically is not well-defined.

To uncover alterations to hippocampal-prefrontal communication underlying impaired executive performance of 6-month-old 3xTgs on the 4CGT, LFPs were recorded from infralimbic layers II/III and V in brain slices following single or paired-pulse electrical stimulation of hippocampal input fibres. This allowed the study of input-output (IO) responses (functional connectivity) and short-term synaptic plasticity, respectively. Response amplitudes to hippocampal input stimulation were significantly reduced in IL layer II/III, whilst the PPR was significantly reduced in layer V, revealing early-stage disruptions to hippocampal-prefrontal connectivity in 3xTg mice that may underlie observed executive dysfunction.

## Results

### 3xTg mice displayed decision-making deficits on the 4-Choice Gambling Task

To assess executive function in an AD context, male 3xTg and control mice were trained to criterion on the 4CGT (Fig. 1), a rodent analogue of the IGT used to assess decision-making in the clinic. Mice began staged training at 2–3 months of age and were fully trained to criterion by 6–7 months of age ($n = 5$ controls, $n = 7$ 3xTgs). The training performance of both groups was similar, with no significant difference in both the overall number of sessions required to reach criterion and the number of sessions required to reach that criterion for each training stage, supporting similar motivation and reinforcement learning abilities between groups. Similarly, during testing, the number of trials completed per session by 3xTg mice was similar to controls, as were the percentages of omissions, premature responses, and number of perseverant responses to the choice location or rest of the grid (Supplementary Fig. 1). Therefore, 3xTg mice were able to learn and perform the task at a similar rate and engagement level to control mice.

Analysis of test sessions revealed no significant effect of session in control (Fig. 2a) or 3xTg (Fig. 2d) mice, nor a significant interaction between session and average choice %. However, there was a significant effect of average choice % in control mice (Fig. 2a, F (3,16) = 24.64; $p < 0.0001$) but not 3xTg mice (Fig. 2d). When sessions were combined (4 sessions per block), the significant effect of average choice % remained (Fig. 2b, F (3, 16) = 29.37; $p < 0.0001$). Control mice showed a clear preference for option P2, choosing it significantly more than other options in the first and second

session blocks (Fig. 2b, $p = 0.0424$ for P2 over P3 in first session block; p < 0.0001 for P2 over P1 and P4, and $p = 0.0003$ for P2 over P3 in second session block). Conversely, 3xTg mice did not choose P2 significantly more than any other option, even following session blocking (Fig. 2e). When combining advantageous versus disadvantageous options (P1 + P2 versus P3 + P4, respectively), control mice showed a clear preference for the two advantageous options (Fig. 2c, F (1,22) = 146.2; $p < 0.0001$) and chose these significantly more than disadvantageous options in all three session blocks ($p = 0.0010$, $p < 0.0001$, $p = 0.0041$) while 3xTgs did not show any preference (Fig. 2f).

Performance on the 4CGT can also be assessed by comparing the net advantageous choice of control and 3xTg mice, as is commonly carried out for human subjects on the IGT in the clinic[24]. This was calculated by subtracting the average % of disadvantageous choices (P3 + P4) chosen over the course of the session from the % of advantageous choices (P1 + P2). Mixed-effects analysis showed a significant effect of genotype on net advantageous choice; control mice had a significantly higher net advantageous choice across individual sessions, averaging 50.93% across 12 sessions, compared to 9.57% for 3xTg mice (Fig. 2g, F (1,10) = 8.740; $p = 0.0144$). When sessions were blocked, this significant effect of genotype remained (Fig. 2h, F (1, 10) = 9.419; $p = 0.0119$). Multiple comparison analysis showed that controls had a significantly higher net advantageous choice in individual session blocks 1 and 2 (Fig. 2h, $p = 0.0220$ and $p = 0.0289$, respectively). Control mice were, therefore, able to establish an advantageous strategy over the first few trials, and then maintain this strategy for subsequent sessions, whereas 3xTgs could not develop an advantageous strategy (Fig. 2g). As other aspects of task performance were similar between groups (Supplementary Fig. 1), the inability of 6-8-month 3xTg mice to perform the 4CGT was most likely due to cognitive dysfunction and deficits in decision-making, as opposed to a lack of motivation, atypical locomotor activity, or an inability to learn the task.

### Altered input/output and paired pulse responses in the prefrontal cortex of 3xTg mice following stimulation of hippocampal inputs

To examine how vHIP-mPFC connectivity may be altered in 3xTg mice, given our observed executive dysfunction, local field responses were recorded in brain slices from layers II/III and V of the IL cortex following stimulation (10 to 100 V intensities) of hippocampal fibres innervating these regions[52] (Fig. 3a). Representative responses for control and 3xTg mice are shown in Fig. 3b, d, respectively. In IL layer II/III, IO curves showed a significant effect of genotype (p < 0.0001) and stimulation strength ($p < 0.0001$) and a significant interaction (Fig. 3c, F (9150) = 3.658; $p = 0.0004$). Pairwise comparisons showed that local field responses in IL layer II/III were significantly lower in 3xTg than control mice at stimulation intensities between 40 and 100 V (Fig. 3c, $p = 0.0432$, $p = 0.0053$, and $p = 0.0007$ for 40 V, 50 V, and 60 V stimulations respectively, p < 0.0001 for 70 V to 100 V stimulations). Additionally, population spikes were more frequently observed in layer II/III of control animals compared to 3xTgs (Fig. 3f, F (3,129) = 92.06; $p < 0.0001$). These observations support a reduced synaptic connectivity between the vHIP and mPFC, and dampened spread of postsynaptic excitation and reduced excitatory postsynaptic potential (EPSP) conversion to spike output within the mPFC of 3xTgs. However, the presence and latency to maximal amplitude for a slower (presumed inhibitory) response component in layer II/III were similar between control and 3xTg LFPs (Fig. 3g, h). Response amplitudes in IL layer V were similar between 3xTgs and control mice (Fig. 3e).

Next, to explore the short-term synaptic plasticity of hippocampal-prefrontal synapses further, responses were recorded following paired-pulse stimulation of innervating hippocampal fibres at 20, 50, 100, 200, and 500 ms intervals. As with IO responses, paired-pulse responses generally exhibited an initial excitatory local field response, followed by a slower presumed inhibitory component. In layer II/III, control mPFC LFPs exhibited mild paired-pulse facilitation (PPF) at all intervals except 20 ms, whereas 3xTg LFPs displayed a PPR very close to 1 (i.e., no facilitation) at all intervals except 20 ms, at which they exhibited paired-pulse depression

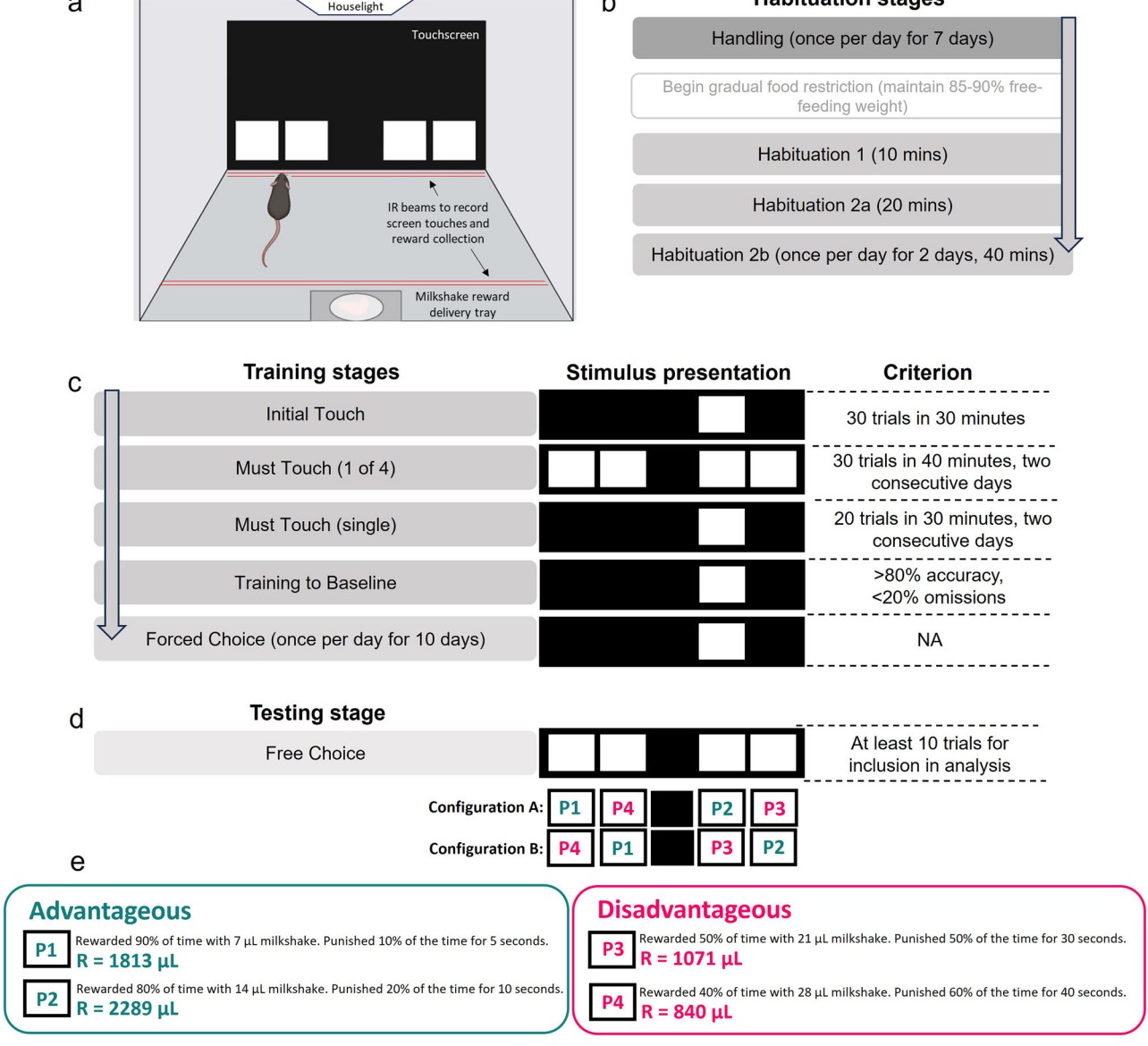

**Fig. 1 | 4-Choice Gambling task (4CGT) habituation, training, and testing procedures. a** Set-up of the touchscreen-operant chamber. **b** Order of 4CGT habituation stages. **c** Order of 4CGT training stages, their associated stimulus presentation and criterion required to pass each stage. **d** The 4CGT testing stage consisted of Free Choice sessions where mice were presented with all four choice stimuli. Two configurations were used, to prevent location bias. **e** Reward/punishment contingencies for each choice location. P1 and P2 were deemed low-risk, advantageous options, whilst P3 and P4 were high-risk, disadvantageous options. This can be deduced by the R-value, which represents the total volume of milkshake received over one 30-minute trial if that choice option is chosen exclusively throughout the trial. Figure 1a was created using BioRender.com.

(PPD). However, 2-way ANOVA analyses of PPRs in layer II/III did not show a significant effect of genotype or interval, nor a significant interaction (Fig. 4a, c). Conversely, in layer V, a significant effect of genotype (F (1,50) = 40.79; $p < 0.0001$) and a significant interaction between genotype and interval was observed (Fig. 4b, d, F (4,50) = 3.756; $p = 0.0095$). Multiple comparisons analysis showed that PPRs at shorter intervals (20, 50, and 100 ms) were significantly reduced in 3xTg compared to control mice (Fig. 4d, $p < 0.0001$, $p = 0.0002$ and $p = 0.0498$, respectively). At all five intervals, 3xTg responses exhibited PPD, indicative of a reduction in short-term synaptic plasticity in the vHIP-mPFC layer V input in 3xTg mice.

## Discussion

Executive dysfunction is a marked feature of dementia and AD; however, our understanding of this symptom realm is lacking, due to relatively little research focus in preclinical models of dementia. Alongside control mice, 6-month-old male 3xTgAD mice were able to train successfully to pre-testing

criterion on the 4CGT. Subsequent testing revealed that, whilst control mice were able to optimise their strategy and choose advantageous (low risk) options a significantly higher proportion of the time than disadvantageous (high risk) options, 3xTg mice did not show preference for advantageous options. Control mice appeared capable of distinguishing the milkshake reward volume between even the two most advantageous options, P1 and P2, evidenced by their significantly higher selection of P2 over other options in most sessions. In contrast, 3xTg mice displayed similar preference for all reward options (increased disadvantageous choices compared to control), suggesting an inability to adapt behaviour based on outcomes (reward prediction error). This profile closely resembles the performance of patients with mild AD on the IGT, the clinical equivalent of the 4CGT[24]. Such poor decision-making on the IGT may arise due to either substantial risk-taking behaviour whilst the task is being undertaken 'under risk' (subjects continuously choose high-risk, high reward options) or the inability to develop an advantageous strategy whilst the task is being performed 'under

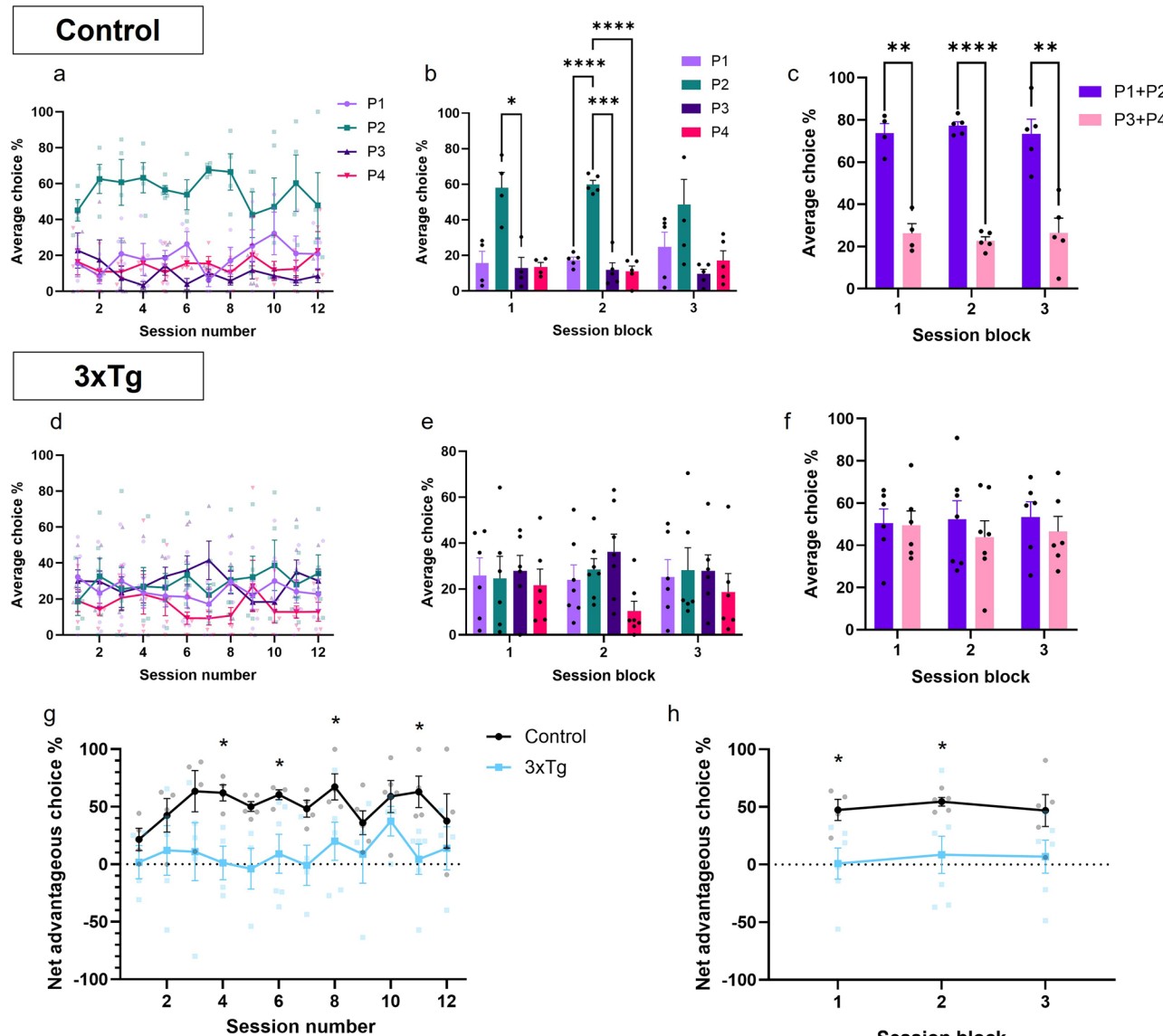

**Fig. 2 | Performance of control and 3xTgAD mice on the 4-Choice Gambling Task. a** Average choice % of control mice by session. **b** The same data as for **a**, but with sessions displayed in blocks of 4. **c** Average choice % of control mice for advantageous (P1 + P2) versus disadvantageous (P3 + P4) options, by session block. **d** Average choice % of 3xTg mice by session. **e** The same data as for **d**, but with sessions displayed in blocks of 4. **f** Average choice % of 3xTg mice for advantageous (P1 + P2) versus disadvantageous (P3 + P4) options, by session block. **g** Net advantageous choice % by individual session. This was calculated by subtracting the average % of disadvantageous choices (P3 + P4) from the average % of advantageous choices (P1 + P2) selected by mice across the session. The higher the value, the better the decision-making and task performance. Control mice had a significantly higher net advantageous choice % than 3xTg mice. **h** The same data as for **g**, but with sessions blocked. Control mice had a significantly higher net advantageous choice % than 3xTg mice. $n = 5$ controls, $n = 7$ 3xTgs. Error bars indicate SEM.

ambiguity' (subjects switch frequently between known advantageous and disadvantageous options without an obvious strategy)[24]. Similarly to IGT results in AD patients, 3xTg mice did not choose predominantly high-risk, high-reward choices, and instead chose a similar percentage of advantageous and disadvantageous options, thereby revealing that poor performance on the task was most likely due to the inability to plan and implement an appropriate strategy. This form of executive dysfunction in AD patients has been attributed to altered function of the dlPFC[21,24,53,54], analogous to the rodent mPFC.

Here, hippocampal input to the mPFC would normally play a vital role in supporting memory encoding by continuously updating task-relevant information to inform decision-making[55]. The hippocampal-prefrontal pathway is, therefore, essential in enabling prefrontal neurons to compute which actions are appropriate in a given spatial context, using memories of that context encoded by the hippocampus[56]. The strengthening of this

pathway, observed by theta-frequency connectivity, has been strongly suggested to enable learning[2], while lesions of the vHIP abolish anticipatory activity in the mPFC, altering impulse control and goal-directed behaviour[57]. The latter results in the inability to utilize task-relevant information for the formation of an appropriate strategy. This choice profile contrasts with examples of risk-taking behaviour where subjects cannot stop choosing high risk, high reward options, as observed in patients with amygdala or ventromedial PFC lesions[58,59]. The present behavioural results, therefore, implicate deficits in vHIP-mPFC connectivity and function in the loss of executive function commonly observed in AD patients[3,12–15].

To determine whether the behavioural outcomes described above were indeed associated with altered hippocampal input to the mPFC, we examined the physiology of this synaptic connection in our 3xTg AD model in vitro. Pyramidal neurons in the mPFC display reduced neuronal activity in other AD mouse models[60,61] and the same phenotype was observed in this

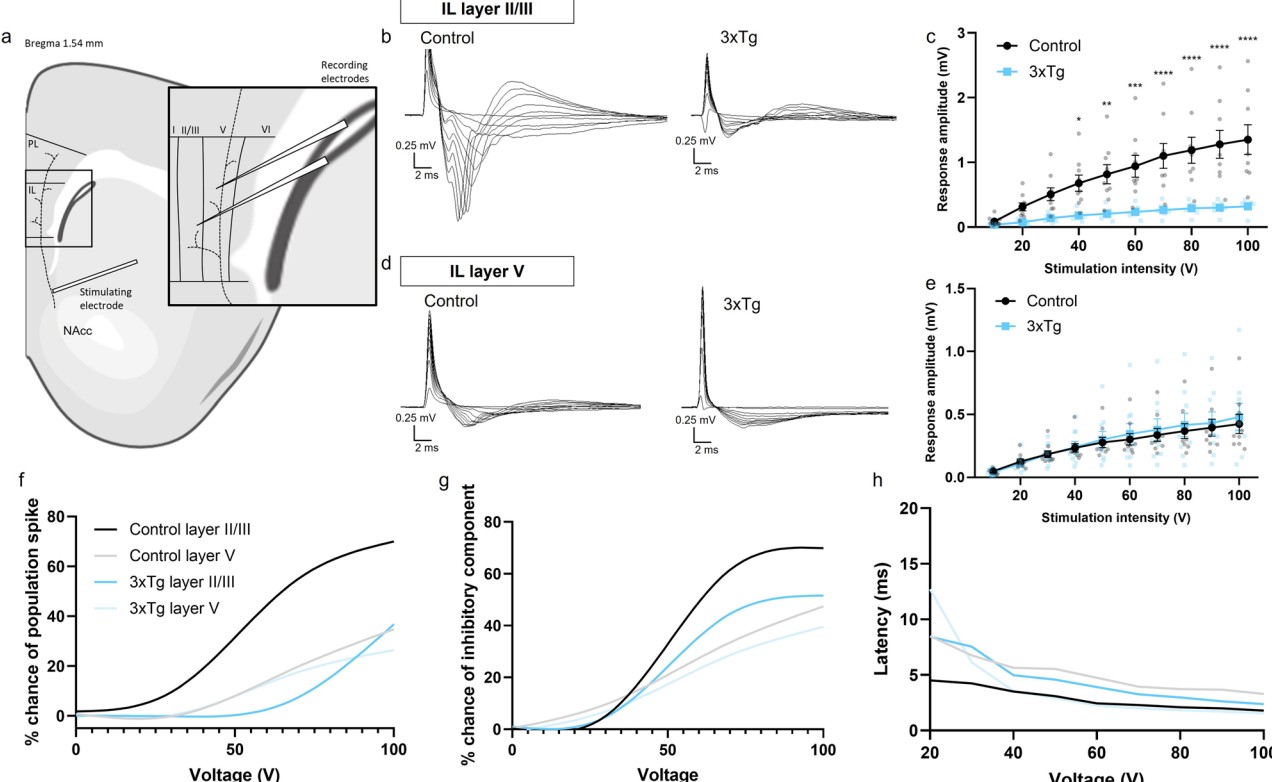

**Fig. 3 | Input/output responses of 3xTgAD and control mice in mPFC layers II/III and V following hippocampal stimulation. a** Schematic showing recording arrangement. The stimulating electrode was placed alongside and dorsal to the nucleus accumbens core (NAcc) to activate hippocampal fibres innervating the infralimbic (IL) cortex of the mPFC. Two recording electrodes were placed in the mPFC, in IL layers II/III and V, respectively, to record local field potentials. **b** Representative traces obtained from IL layer II/III of control (left) and 3xTg (right) slices at 10–100 V stimulating strengths. **c** Input/output curve comparing response amplitudes in layer II/III of control versus 3xTg mice. **d** Representative traces obtained from IL layer V of control (left) and 3xTg (right) slices. **e** Input/output curve comparing response amplitudes in layer V of control versus 3xTg mice. **f** % chance of observing a population spike in control versus 3xTg mice in layers II/III and V. **g** % chance of observing a late inhibitory postsynaptic potential (IPSP) response in control versus 3xTg mice IL layers II/III and V. **h** Latency to maximum response amplitude in control versus 3xTg mice IL layers II/III and V. Data from one slice per animal, $n = 9$ controls and $n = 8$ 3xTgs. Error bars indicate SEM. Figure 3a was created using BioRender.com.

study, specifically in 3xTg prefrontal neurons receiving direct hippocampal input. Prefrontal local field responses to hippocampal fibre stimulation display an initial excitatory phase, reflecting positive ion influx (known as the excitatory postsynaptic current, EPSC) that leads to fast AMPA-receptor and slow NMDA-receptor activation in the form of an EPSP[9,62,63], sometimes succeeded by a population spike. The initial excitatory response is followed by an inhibitory postsynaptic potential (IPSP), as a result of GABA_A and GABA_B receptor activation, which leads to chloride ion influx and potassium ion efflux (underlying the inhibitory postsynaptic current, IPSC)[9,64]. Therefore, IO response amplitudes depend on the level of activation of both glutamatergic and GABAergic receptors. Consequently, reductions in the amplitudes of these responses may occur as a direct result of either increased GABAergic inhibitory transmission, or attenuated glutamate receptor expression and synaptic transmission (reduced glutamatergic binding at the postsynaptic membrane). Reduced expression of vesicular glutamate transporter (VGLUT) 1 and VGLUT2 has been observed in prefrontal regions of AD patients, correlating with cognitive decline[36] and reduced *Gad1* (encoding for the glutamate decarboxylase 1 protein) expression has previously been observed in the prefrontal cortex of AD patients[35], which inevitably disrupts the glutamate-glycine-GABA cycle. Reduced expression of GAD1 in the dlPFC has been suggested to contribute towards E-I network dysfunction in schizophrenia[65], so it may be that a similar mechanism is occurring in the dlPFC of AD patients. Furthermore, D1 receptors have been shown to colocalise with glutamate receptors in PFC pyramidal cells[66], and both dopamine receptors 1 and 2 (D1 and D2 receptors) directly modulate glutamate receptor function and transmission

and synaptic plasticity[67–70]. Expression of D1 and D2 receptors in the PFC are considered critical for behavioural flexibility[71], reward-driven behaviours[72], and learning and memory processes dependent on synaptic plasticity[2,73,74] and dopamine is reported to modulate glutamatergic activity associated with hippocampal-prefrontal plasticity[75,76]. Interestingly, the application of a D2 receptor antagonist has been shown to improve performance on the 4CGT in male rats[25], suggesting that overactivation of D2 receptors may also contribute towards impaired decision-making.

In this study, following single pulse stimulation of hippocampal input fibres in 3xTgs, a significant reduction in mPFC local field response amplitude was observed compared to controls in mPFC layer II/III, but not layer V. Additionally, mPFC LFPs were less likely to display a population spike in layer II/III of 3xTg mice. Therefore, by recording LFPs in the mPFC following hippocampal fibre stimulation, the connectivity between the hippocampus and layer II/III mPFC of 3xTg mice was shown to be compromised, possibly due to a general increase in inhibition, and/or a reduction in excitation in the mPFC. Local mPFC circuitry involving interneurons has been shown to be layer specific; layer II/III interneurons receiving hippocampal input predominantly express vasoactive intestinal polypeptide (VIP), while those in layer V express parvalbumin (PV), somatostatin (SOM) or cholecystokinin (CCK)[77–80]. Layer V interneuron subtypes have been suggested to contribute towards feedforward and feedback inhibition by synapsing directly with excitatory pyramidal cells in layer V[77,81,82]. Conversely, VIP-expressing interneurons inhibit other interneuron subtypes in layer II/III[8], thus, leading to disinhibition. Observed layer-specific differences in the response amplitude may, therefore, be

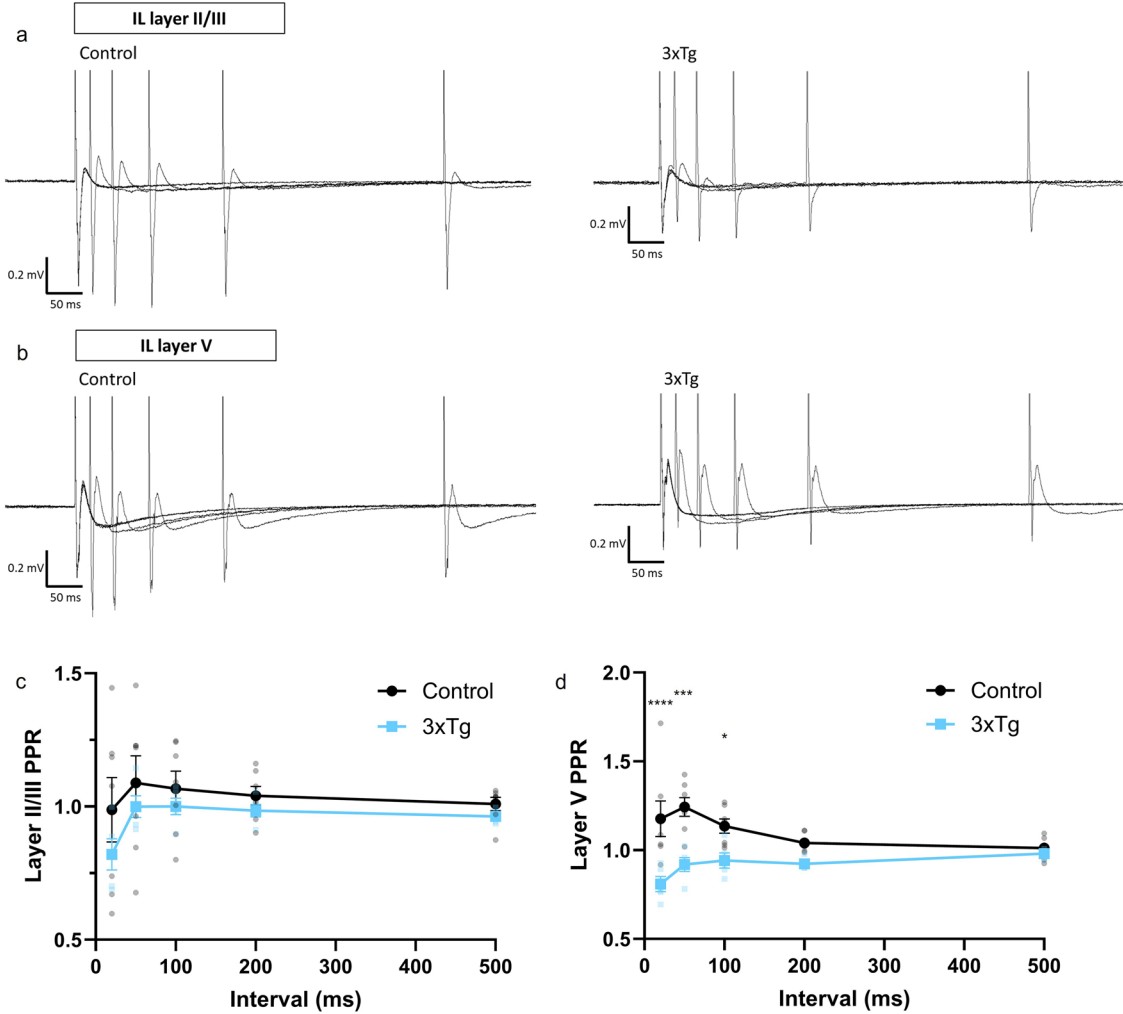

**Fig. 4 | Responses of 3xTg and control mice in mPFC layers II/III and V following 100 V hippocampal paired pulse stimulation at 20, 50, 100, 200, and 500 ms.** **a** Representative traces from IL layer II/III of control (left) and 3xTg (right) slices. **b** Representative traces from IL layer V of control (left) and 3xTg (right) slices. **c** Quantification of paired pulse ratios at each interval in layer II/III of 3xTgs and controls. **d** Quantification of paired pulse ratios at each interval in layer V of 3xTgs and controls. Data from one slice per animal, $n = 9$ controls and $n = 5$ 3xTgs. Error bars indicate SEM.

associated with differences in local interneuron circuitry. Previous studies have reported changes to firing properties of VIP-positive interneurons in the hippocampus in young 3xTg mice[83], and during ageing[84], although whether alterations to VIP-positive interneurons also occur in layer II/III of the mPFC, and how this may underlie synaptic alterations, have yet to be determined.

Short-term plasticity changes in the vHIP-mPFC pathway were explored via paired pulse recordings. PPF in prelimbic and infralimbic regions following hippocampal paired-pulse stimulation at intervals between 50 and 150 ms have previously been observed, with largest PPF reportedly occurring at an interval of 50-75ms[85]. A PPR less than 1 indicates that the first pulse is relatively larger than the second, and this is caused by increased vesicle release probability and subsequent vesicle depletion associated with PPD and synaptic weakening. A lower vesicle release probability results in a PPR larger than 1, indicative of PPF and synaptic strengthening[86]. Presynaptic vesicle exocytosis is triggered by the influx of calcium, which initiates the activation of vesicle release machinery, namely synaptotagmins, complexins and SNARE proteins[44]. Therefore, changes to release probability can be explained by variations in the intracellular accumulation of residual calcium or functioning of vesicle release machinery.

In this study, significant reductions in the PPR were observed in mPFC LFPs of 3xTg mice compared to controls in layer V, but not layer II/III. It is possible that the lack of change in layer II/III is a consequence of the large

reduction in response amplitudes following stimulation of excitatory hippocampal input in 3xTgs, which becomes insufficient to generate the levels of activation and recruitment of inhibition required to reveal changes to the PPR. In layer V, PPRs were significantly lower in 3xTgs, and exhibited PPD as opposed to the PPF observed in controls animals. These changes propose alterations to the vesicle release probability and/or residual calcium concentration of vHIP-mPFC synapses. Dysregulation of calcium ion dynamics has been linked with the accumulation of Aβ and is widely suggested to contribute towards aberrant synaptic signalling underlying neurodegeneration[47–50]. An increased concentration of cytosolic calcium has also been linked to upregulated intraneuronal amyloid production, leading to cell death[87]. Furthermore, observed effects were particularly notable at shorter intervals (25, 50, and 100 ms), possibly indicating additional upregulation of inhibition mediated by GABA$_A$ receptors.

These results suggest that altered neuromodulation of glutamate and GABA in the mPFC, as well as disruptions to calcium ion transport and vesicle release probability, occur in early stages of the disease, preceding the appearance of Aβ plaques and tau tangles. It is possible that early changes to glutamatergic activity in layer II/III drive postsynaptic reductions in effective connectivity in the hippocampal-prefrontal pathway, which is counteracted by altered calcium transport at the presynaptic membrane, leading to increased vesicle release and thus PPD of hippocampal-prefrontal synapses in layer V. Changes to dopaminergic modulation of this pathway,

specifically abnormal activity of D1 and D2 receptors, are likely to contribute towards these observations[1,70,88,89].

Future studies may shed light on the precise neuronal type and the intricate neural networks, including dopaminergic, glutaminergic, and GABAergic pathways connecting to the mPFC region, that may be impaired in 3xTg mice. Investigating these questions using techniques such as patch-clamp electrophysiology or single-cell RNA sequencing will enable the determination of more precise molecular mechanisms that underpin alterations to this pathway in 3xTg mice. If these can be identified, it may be possible to selectively activate or inhibit specific neuronal subtypes during behavioural testing to directly correlate alterations in the hippocampal-prefrontal pathway with executive dysfunction. This will enable a more detailed understanding of pathophysiology underlying early AD stages, prior to the overt accumulation of Aβ plaques and tau neurofibrillary tangles. This represents a critical stage for improving diagnostics and disease-rectifying interventions.

## Conclusions

These results show that 3xTgAD mice can be trained to pre-test criterion on the touchscreen operant 4-Choice Gambling Task, and that this rodent version of the clinically utilized Iowa Gambling Task can be used successfully to observe executive dysfunction in an AD context. The pre-test performance of 3xTg and control mice was similar, suggesting that differences in motivation and reinforcement learning abilities were not present between groups. However, at test 3xTg mice displayed poorer performance on the task than matched controls, indicative of deficits in decision-making that are symptomatic of Alzheimer's disease. Alongside these results, altered mPFC responses to activation of hippocampal input fibres in 3xTg mice may provide an explanation for observed executive dysfunction. It must be noted, however, that final sample sizes for behavioural testing were somewhat limited, and while the present results strongly support decision-making deficits in 3xTg mice on the task, further attempts using the 3xTg and other AD models to replicate and extend these findings would be valuable.

## Methods
### Animals

Male triple transgenic (3xTgAD; $n = 9$) mice carrying APP$_{SWE}$, PS1$_{M146V}$ and Tau$_{P301L}$ transgenes, and matched controls ($n = 9$) of the same background strain (C57/129sv) were 2-3 months old at the start of behavioural training. These sample sizes were based on: (a) pilot data from $n = 10$ age-matched female C57BL/6 mice demonstrating a significant preference for optimal choices during 4CGT testing (Supplementary Fig. 2e–g); and (b) age-matched transgenic mouse availability. At the start of behavioural testing, $n = 7$ 3xTg and $n = 5$ control mice were 6-7 months old and had met pre-testing criteria. Mice that had not reached pre-testing criteria ($n = 2$ 3xTg mice and $n = 4$ control mice) were removed from behavioural testing analyses. Failure to meet criteria typically occurred due to an insufficient number of completed trials, suggesting a lack of motivation to perform the task for these animals. Indeed, when compared with C57 mice used in pilot studies, 3xTg 129sv control mice made significantly more omissions (Supplementary Fig. 2b) and completed significantly fewer trials per session (Supplementary Fig. 2d). However, the total number of sessions required to reach criterion was not significantly different to that of C57s (Supplementary Fig. 2a), and testing performance was similar across strains (Supplementary Fig. 2e-l). Notably, 129sv mice completed the task slightly more efficiently, as they demonstrated a strong choice preference for the most optimal choice; P2 (Supplementary Fig. 2i), whilst C57 mice chose advantageous choices P1 and P2 a similar amount (Supplementary Fig. 2f).

For electrophysiology experiments, 3xTg ($n = 8$ and $n = 5$ for IO and PPR experiments, respectively) and control ($n = 9$) mice were 6 months of age, an approximate age-match to behaviourally tested mice. Mice were bred from homozygous pairs (Jackson Labs) in the University of Manchester Biological Services Facility. Genotyping of offspring was confirmed by Transnetyx using PCR amplification of tissue samples collected from ear

punches. Mice were housed five per cage and maintained on a 12-hour light/dark cycle (lights off at 7 pm). Mice had access to food and water ad libitum, except during behavioural training and testing, when they were food restricted to encourage behavioural performance whilst maintaining body weight no lower than 85–90% of free feeding animals. All experimental procedures were performed under a Home Office UK project licence in accordance with the Animals (Scientific Procedures) Act 1986, and approval by the University of Manchester Animal Welfare and Ethical Review Body (AWERB). We have complied with all relevant ethical regulations for animal use.

### Behavioural testing

Touchscreen-operant chambers (Campden Instruments Ltd., UK), as previously described[90–92], consisted of a trapezoidal floor surrounded by black Perspex walls, under which there was a waste-tray. The longest edge of the chamber contained the touchscreen, in front of which was a black Perspex mask containing five response windows. These windows corresponded to squares which would light up on the screen, with all but the middle square being used during 4 CGT training and testing. The opposite side of the chamber contained a food delivery magazine and a light. Infra-red beams to measure task performance were installed in front of the touchscreen and the food reward trough (Fig. 1a). Eight chambers were used simultaneously within individual light- and sound-attenuating lockable boxes. These boxes also contained a house light, tone generator, ventilating fan, and camera. A peristaltic pump was used to deliver behavioural reinforcer (Yazoo strawberry milkshake, FrieslandCampina, Ltd., UK) into the food delivery magazine. ABET II software (Campden Instruments Ltd, UK) was used to control each touchscreen chamber. Training and testing protocols were performed according to the Mouse Touch 4C-GT ABET II Manual V1.2 (Campden Instruments Ltd., UK).

Prior to behavioural training, mice were habituated to experimenter handling every day for one week. They were then gradually food restricted for four days prior to the first training day to increase motivation for milkshake reward. Food restriction was continued throughout training and testing stages, to maintain motivation. Each mouse was first placed inside an experimental chamber for 10 min to allow habituation to the apparatus. On the second day of habituation, mice were left for 20 min, and milkshake reward (7 μl) was dispensed each time the mouse poked its head into the milkshake delivery tray. This was repeated on the third and fourth habituation days, which were 40 min long. By the fourth habituation day, all mice were able to successfully find and collect milkshake reward. Habituation stages are summarised in Fig. 1b.

Behavioural training (Fig. 1c) began immediately following habituation stages. Mice were given one daily training session, five days a week. The first stage was Initial Touch training, consisting of one white square being pseudo-randomly displayed in one of the four possible locations for 30 s. The square was then removed and 7 μl milkshake delivered (accompanied by a tone). If the square was nose poked during its display 20 μl milkshake was rewarded, encouraging touches to the stimuli. A new trial would be initiated once the mouse consumed the milkshake reward. Criterion at this stage was the completion of 30 trials in 30 min. The next stage required mice to touch the screen to receive milkshake reward; all four possible square locations were lit up and mice were rewarded with 7 μl milkshake for touching any 1 of the 4 squares. Criterion for this stage was 40 successful touches to the light cues within 30 min, on two consecutive days. The next stage was the same, except a single square only was pseudo-randomly presented across the four possible positions on a given trial, and milkshake dispensed only following touches to that target. Following the successful completion of 20 trials in 30 min on two consecutive days, mice were moved onto the final training stage. This was the same as the previous stage, except a time limit was imposed on the stimulus presentation, and mice were punished (luminance inverted for 5 s) for touches to incorrect/blank locations. This Training to Baseline stage contained up to 4 sessions, with the stimulus duration for each session lower than the one before (37, 21, 13, and 10 s, respectively). Moving to the next session required mice to finish the previous

session with >80% accuracy of touches to the correct stimuli and <20% trial omissions. Inclusion of this training stage is not deemed necessary[32], but the first Training to Baseline training stage (37-s stimulus duration) was included in this study to verify that mice could execute the task. For each training and testing stage, the number of trials completed, % accuracy and % omissions were recorded.

Finally, following training and prior to the Free Choice version of the task, mice were put through ten Forced Choice sessions where just one stimulus position was illuminated at once, to familiarise them completely with the reward/punishment possibilities for each position. Once they had successfully moved through all Forced Choice sessions, animals were tested on the Free Choice task. For both Forced Choice and Free Choice sessions, two spatial configurations of 4 positions were used to counterbalance possible location biases. Therefore, half the mice faced an order of choices of (from left to right); P1, P4, P2, P3 (configuration A), and the other half P4, P1, P3, P2 (configuration B, Fig. 1d).

The Free Choice task began with 7 μl milkshake delivery, accompanied by a tone. Once the mouse nose poked the delivery magazine, it was presented with white square stimuli in positions 1, 2, 4, and 5. Mice had to touch one square, at which point they would receive either a milkshake reward or a time-out (the image would flash at a frequency of 5 Hz and chamber lights were inverted). Each square was associated with a different volume of milkshake reward, a different probability of receiving this reward instead of a time-out, and a different length of time-out if this occurred instead. Touches to squares with lower volumes of reward were also associated with a greater chance of reward, and shorter time-out length if the reward was not received. For example, touches to position 2 (P2) dispensed 14 μl of milkshake 80% of the time, while 20% of touches resulted in a 10 second time-out. Alternatively, touches to position 4 (P4) dispensed 28 μl of milkshake 40% of the time, and time-outs occurred for 60% of touches and were a much longer 40 s. A summary of all 4 positions' reward/punishment contingencies can be seen in Fig. 1e. The order of best to worst choice in terms of highest long-term reward pay-off was calculated by comparing the maximum milkshake reward for each choice if that choice was chosen exclusively throughout the 30-min task. This is referred to as the R value, with a higher value correlating with a larger volume of reward for that choice. The order of choices with highest to lowest R number was: P2 > P1 > P3 > P4. Therefore, the optimal volume of reward per unit time is achieved when the square at P2 is continuously chosen, and high-risk, high-reward options (P3 and P4) are avoided.

## Electrophysiology

Mice were euthanised by cervical dislocation and brains quickly removed into ice-cold sucrose Krebs solution, which contained (in mM): 26 NaHCO3, 2 KCl, 1.25 KH2PO4, 10 MgSO4, 0.5 CaCl2, 10 D-glucose and 202 sucrose (290–295 mOsm) and was continuously bubbled throughout slice preparation with 95% $O_2$/5% $CO_2$. Coronal slices (400 μM) were cut using the Vibroslice HA752 vibratome (Campden Instruments Ltd) in the same ice-cold solution, before they were transferred into holding chambers at 37 °C containing artificial cerebrospinal fluid (aCSF). The aCSF solution consisted of (in mM): 124 NaCl, 26 NaHCO3, 2 KCl, 1.25 KH2PO4, 1.5 MgSO4, 1.5 CaCl2 and 10 D-glucose (290–295 mOsm) and was continuously bubbled with 95% $O_2$/5% $CO_2$ throughout slice recovery and recordings. Slices were left to recover for at least 1 h.

Following recovery, slices were placed into a recording chamber, which was constantly perfused with 32 °C aCSF at a flow rate of 0.7 ml/min. The recording chamber beneath the brain slice was filled with water, which was also bubbled with 95% $O_2$/5% $CO_2$ and heated to 32 °C, ensuring a humidified, oxygen-rich atmosphere surrounded the slice. Once the slice had been placed in the recording area, a stimulating electrode was placed in the target area. Because the aim was to stimulate hippocampal fibres innervating the mPFC, the stimulating electrode was placed around 1 cm ventral to infralimbic cortex layers V/VI, slightly dorsal to the nucleus accumbens core, which is the location these fibres have been reported to run through just prior to innervating the mPFC[52]. The stimulating electrode was made by

twisting together two wires made of Teflon-insulated stainless steel, approximately 125 μm in diameter with bare ends (Advent RM, UK). Glass recording electrodes were pulled to a tip diameter of 10-15 μm with an impedance c.1 MΩ. Two recording electrodes were filled with aCSF solution and individually placed in layers II/III and V of the infralimbic cortex. These were the recording areas of choice due to the relative preservation of hippocampal fibre connectivity in the slice preparation compared to the more dorsal prelimbic cortex[52].

Local field potentials in response to constant 10–100 V stimulation were recorded from layers II/III and layer V, followed by paired-pulse responses to 100 V stimulation with intervals of 20, 50, 100, 200, and 500 ms. The maximal 100 V stimulation was used to ensure optimal activation of local mPFC circuitry, enabling clear observation of any differences in response amplitudes between 3xTgs and controls. Five paired-pulse response traces were obtained per interval. Traces were subsequently averaged, and amplitudes and latencies of responses measured using Signal software (version 5; CED, UK).

## Statistics

4CGT and electrophysiology data were first tested for normality using the Shapiro-Wilk test, and equal variance using the F test. Unless stated, normally distributed 4CGT and electrophysiological data were analysed using a mixed-effects analysis or 2-way ANOVA respectively, with multiple comparisons (Bonferroni post hoc analysis). All testing was performed with Prism software (version 9.5.1, GraphPad UK). For behavioural data, if a mouse completed less than 10 trials over a session, this individual performance was excluded from the grouped analysis. Control and 3xTg performance were only analysed for $n = 4$ animals or more per group (i.e., 4 or more animals had performed 10 trials or more over the session).

## Reporting summary

Further information on research design is available in the Nature Portfolio Reporting Summary linked to this article.

## Data availability

The source data behind the graphs in the paper can be found in Supplementary Data 1. Any further datasets generated during and/or analysed during the current study are available from the corresponding author on reasonable request.

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

## Acknowledgements
We would like to thank Mr Matthew Burgess for his expertise in helping set up behavioural testing chambers. This study was supported by the A*STAR Research Attachment (ARAP) PhD programme.

## Author contributions
J.G. conceptualised the project; G.C., J.T. and J.G. designed the experiments; G.C. performed experiments and analyses relating to behavioural and electrophysiological data; G.C., L.Y.T., J.G. and J.T. interpreted the results; G.C. wrote the original draft of the manuscript; all authors helped critically review the manuscript; J.G., J.T. and J.S. provided supervision over the project.

## Competing interests
The authors declare no conflicts of interest with respect to the research, authorship, or publication of this article.
