## [Transparent Peer Review file · Communications Biology]

Executive dysfunction is associated with altered hippocampal-prefrontal functional connectivity in male 3xTg Alzheimer's model mice

Corresponding Author: Dr John Gigg

Version 0:

Reviewer comments:

Reviewer #1

(Remarks to the Author)

For the paper titled "Executive dysfunction is associated with altered hippocampal-prefrontal functional connectivity in male 3xTg Alzheimer's model mice", the authors utilized the 3xTg mouse model to examine decision making as a model of executive function that is reliant on the ventral hippocampus connectivity to the medial prefrontal cortex. They utilized a fairly difficult to learn task, the 4 choice gambling task to perform this in mid-aged mice, combined with field recordings and paired pulse recordings. They followed this with bulk RNA-seq in the mPFC.

Major concerns:

1. the 4 choice gambling task appears too difficult for the 3xTg to learn. They started with n=9 per group, but removed over 20% of the controls and 40% of the 3xTg mice for failure to meet "pre-testing criteria". This loss meant they had very low numbers for behavioral testing and calls into question whether this was sufficient for true statistical analysis. Power calculations should be performed on this cohort and additional animals added for validity. Current numbers are not sufficient for behavioral analysis.
2. The RNA-seq was only performed on n=4 mice in the mPFC only. Again, these numbers are low and it would be interesting to see if the ventral hippocampus showed differential results. Further, the mice were cervically dislocated prior to RNAseq instead of perfusion, to remove peripheral contaminants including macrophages, which would be expected to skew the data in the 3xTg mice, especially under stress conditions, such as caloric restriction for behavioral analysis.
3. The methods are not adequately described for RNA-seq. There is no mention of RNase free conditions or how the bioinformatic analysis was performed. The authors need to get these details from their outsource for validity and replication.
4. The Results on the RNA-seq analysis was vague, with no details on the total number of dysregulated genes. Further, "no descriptive terms" (lines 171-172) for 3MO data indicates that they did not perform the analysis on enough animals, as 3xTg start to have neuroinflammation prior to onset of pathology, which per the introduction start at ~4MO (lines 53-54). Further, the authors assume changes in pathways based on limited (7) DEGs they show in figure 5 for the early changes, but this is not clear, since the authors did not include the total DEGs. Figure 6 only shows a heatmap of the synaptic and neurotransmitter function dysregulation, not the full list, nor are the genes listed legible for follow-up. Suggest the authors include supplemental tables at a minimum of the data or share via online sharing platforms such as GEO or Synapse.org
5. The introduction details on the 3xTg mice is minimal, especially when the authors reference a "lengthy prodromal period" (lines 54-57).
6. Reports of the 3xTg have indicated there has been drift in the phenotype of this mouse line. No amyloid or NTF staining was done to show these animals examined had pathology at either timepoint measured by RNA-seq or electrophysiology.

Reviewer #2

(Remarks to the Author)

This study provides evidence of functional connectivity alterations in the hippocampal-infra-limbic pathway which are assumed to be at the origin of the impairment in executive functions shown by 3xTg AD mice in the 4-Choice Gambling Task (4CGT). The point, however, is that the electrophysiological and gene expression alterations they detect are undoubtedly "consistent" with their hypothesis, but do not actually demonstrate a "causal" link between behavioral, neural and genetic abnormalities. This is due to the fact that these measurements are carried out in non-trained animals and, hence, cannot inform on how plasticity and gene expression abnormalities develop when brain circuits and regions are behaviorally activated; on the other, and they do verify whether modulating hippocampus-prefrontal activity (e.g. via DREADDs) rescue behavioral deficits.

This being said, the report that 3xTgAD mice are unable to implement one advantageous strategy to collect milk in the 4CGT, together with the detection of layer-specific infra-limbic cortex plastic alterations (response amplitude and paired-pulse ratio) following hippocampal fibers stimulation are innovative and could deserve publication although major manuscript revision and additional experiments are necessary.

Major points

Behavior

Main point

The authors' statement is that, differently from control mice, AD mice fail to implement an advantageous strategy. However, although both P1 and P2 are advantageous, only P2 is preferentially chosen by control mice. Thus, as far as P1 is concerned, AD mice behave as do control mice, which means that they do not have a global executive dysfunction. It is therefore mandatory to understand why P2 is preferentially chosen by control mice compared to P1, and why AD mice do not do the same, knowing that the main difference between P1 and P2 is not in the number of rewarded or punished responses, but in the quantity of milk delivered. Thus, even though the authors claim that motivation is unaltered in AD mice, being the quantity of milk delivered apparently irrelevant for AD mice, some difference between genotypes in reward systems cannot be excluded. Accordingly, the interpretation of behavioral differences between genotypes needs to be re-oriented considering the respective motivational/rewarding values of P1 vs P2, and mice have to be tested in another task assessing executive functions with a negative reinforcement (or suppression of reinforcement, e.g. extinction) to estimate in which conditions their executive functions are impaired.

Minor points

The 4CGT is a complex task which apparently engages mice for months. Despite the description of the experimental procedure in both the Results and Methods sections, it is unclear what happens between 3 and 6 months (Mice began staged training at 2-3 months of age and were fully trained to criterion by 6-7 months of age (n= 5 controls, n= 7 3xTgs). Does this mean that the training phase goes on from 3 to 6/7 months? That there is some pause during the training stage? In other words, how many weeks training consists in, and at what age weekly sessions are administered?

Histograms and curves summing up advantageous (P1+P2) vs disadvantageous (P3+P4) choices have to be removed from Figure 2 given that P1 choices do not vary between genotypes so that these data "hide" the fact that AD mice show regular advantageous P1 choices.

Electrophysiology

The authors should speculate more on why the deficit in input/output responses is layer II/III specific while the deficit in pair-pulse response is layer V- specific. More references to the anatomy of ventral hippocampal projections to, and anatomical properties of layer II/III vs Layer V in, infra-limbic (IL) cortex are needed. Also the dysregulation of E/I balance needs to be supported by data showing how interneurons contribute. For example, the statement that "Observed layer-specific differences in the response amplitude may, therefore, be associated with differences in local interneuron circuitry" absolutely requires to perform vasoactive intestinal polypeptide (VIP) and parvalbumin (PV), parvalbumin immunohistochemistry in IL slices.

Gene expression alterations

This experimental part is carefully executed but largely non-specific to the manuscript topic for two reasons. First, changes in gene expression are measured in global mPFC extracts and hence do not inform on how these genes are dysregulated in the infra-limbic cortex. Second, the detected gene alterations are not specific to AD (see Tripathi, M.K et al. Sci Rep 14, 10 (2024). <https://doi.org/10.1038/s41598-023-50248-4>). In my opinion these data could be suppressed from the present manuscript to be the object of an independent publication.

General comments

All sections of the manuscript are too long and need to be shortened.

The discussion largely consists in simply commenting the findings without addressing their main interest (does the 4CGT task reveal reward system alterations or executive function deficits in AD mice?) or their translational potential to diagnose or cure AD.

Version 1:

Reviewer comments:

Reviewer #1

(Remarks to the Author)

The Reviewer appreciates the gambling task was a difficult behavioral test with a long and involved training paradigm. However, there are several issues that raise concerns over rigor. Why did twice as many 'control' subjects fail to reach criteria than transgenic mice? This is worrisome and is not reasonably explained in the manuscript. Further, the Reviewer strongly disagrees that differences between conditions in the end result obviate power calculations. This is not rigorous. Most behavioral tests require much larger cohorts, especially with the difficulty in learning the task increasing, not decreasing the number of cases required to validate behavior. At a minimum, the number of dropped cases needs to be validated by independent means. A second behavioral test for executive function could be utilized but at least one well-powered method is necessary to draw conclusions for degeneration of executive functional behavior prior to overt pathology.

The reviewer agrees that the RNAseq data could be eliminated from the manuscript.

Reviewer #2

(Remarks to the Author)

The ms has been revised by taking the reviewers points adequately. This new version is about ready for publication. Just two minors points remain that can be easily addressed.

1) It is necessary to include a small sentence in the discussion to explain why, among the two advantageous choices, control mice choose considerably more P2 than P1, making it the advantageous P1 chosen with the same frequency as the two disadvantageous P3 and P4. Maybe the difference in the amount of milk delivered (the double in P2 than in P) is the crucial factor which makes P2 overshadowing P1. Anyway, this huge difference between the two advantageous choices emerges so clearly in Fig2b that it is impossible not to comment it.

2) The discussion includes sub-headings for electrophysiological data which undoubtedly facilitate the reading. I suggest to follow the same formula all along the discussion with a sub-heading "Executive functions" at line 174, and "Conclusions" at line 297.

We thank both reviewers for their responses and feedback on the manuscript and have addressed these comments in our revised manuscript. We have highlighted changes to the manuscript in yellow and have provided line numbers in our responses to reviewers, to ensure changes are easy to identify.

Reviewer #1 (Remarks to the Author):

Major concerns:

1. the 4 choice gambling task appears too difficult for the 3xTg to learn. They started with n=9 per group, but removed over 20% of the controls and 40% of the 3xTg mice for failure to meet "pre-testing criteria". This loss meant they had very low numbers for behavioral testing and calls into question whether this was sufficient for true statistical analysis. Power calculations should be performed on this cohort and additional animals added for validity. Current numbers are not sufficient for behavioral analysis.

We agree that our final sample sizes after exclusion were smaller than we would have preferred. However, the effect size is clear and large in the controls (Fig2a-c) and the variance is relatively small for both groups. There is no obvious part of the data where we are overlooking an effect due to a sample size issue – so it is not clear where a power analysis would be beneficial. Thus, we feel the present sample sizes are adequate and the statistical analyses and results support the observed patterns in the data. Our use of a repeated measures ANOVA design helps improve statistical power with smaller sample sizes of course. Additionally, we'd like to highlight that more controls did not reach criteria on the task (around 40%) than 3xTg mice (around 20%), as we believe it is important to note that 3xTg mice were not worse at learning the task than control mice.

2. The RNA-seq was only performed on n=4 mice in the mPFC only. Again, these numbers are low and it would be interesting to see if the ventral hippocampus showed differential results. Further, the mice were cervically dislocated prior to RNAseq instead of perfusion, to remove peripheral contaminants including macrophages, which would be expected to skew the data in the 3xTg mice, especially under stress conditions, such as caloric restriction for behavioral analysis.

See below

3. The methods are not adequately described for RNA-seq. There is no mention of RNase free conditions or how the bioinformatic analysis was performed. The authors need to get these details from their outsource for validity and replication.

See below

4. The Results on the RNA-seq analysis was vague, with no details on the total number of dysregulated genes. Further, "no descriptive terms" (lines 171-172) for 3MO data indicates that they did not perform the analysis on enough animals, as 3xTg start to have neuroinflammation prior to onset of pathology, which per the introduction start at ~4MO (lines 53-54). Further, the authors assume changes in pathways based on limited (7) DEGs they show in figure 5 for the early changes, but this is not clear, since the authors did not include the total DEGs. Figure 6 only shows a heatmap of the synaptic and neurotransmitter function dysregulation, not the full list, nor are the genes listed legible for follow-up. Suggest the authors include supplemental

tables at a minimum of the data or share via online sharing platforms such as GEO or Synapse.org

We agree with the comments from both reviewers on the scope of the RNA-seq data and have removed these from the manuscript. We feel the paper still stands with the remaining behaviour and electrophysiological analyses.

5. The introduction details on the 3xTg mice is minimal, especially when the authors reference a "lengthy prodromal period" (lines 54-57).

We have added some further description to clarify that the lengthy prodromal period refers to the fact that 3xTg mice do not present with extracellular amyloid beta plaques or intracellular tau neurofibrillary tangles until 12 months of age (lines 56-58), thus, the ages tested here relate to early-stage AD in human patients.

6. Reports of the 3xTg have indicated there has been drift in the phenotype of this mouse line. No amyloid or NTF staining was done to show these animals examined had pathology at either timepoint measured by RNA-seq or electrophysiology.

The mice were first generation from a newly imported JAX-derived cohort. Thus, we consider it unlikely that the phenotype was not fully present. We also confirmed the presence of transgenes in the F1 generation by PCR (TransNetyx) before experiments started.

Reviewer #2 (Remarks to the Author):

This study provides evidence of functional connectivity alterations in the hippocampal-intra-limbic pathway which are assumed to be at the origin of the impairment in executive functions shown by 3xTg AD mice in the 4-Choice Gambling Task (4CGT). The point, however, is that the electrophysiological and gene expression alterations they detect are undoubtedly "consistent" with their hypothesis, but do not actually demonstrate a "causal" link between behavioral, neural and genetic abnormalities. This is due to the fact that these measurements are carried out in non-trained animals and, hence, cannot inform on how plasticity and gene expression abnormalities develop when brain circuits and regions are behaviorally activated; on the other, and they do verify whether modulating hippocampus-prefrontal activity (e.g. via DREADDs) rescue behavioral deficits.

This being said, the report that 3xTgAD mice are unable to implement one advantageous strategy to collect milk in the 4CGT, together with the detection of layer-specific infra-limbic cortex plastic alterations (response amplitude and paired-pulse ratio) following hippocampal fibers stimulation are innovative and could deserve publication although major manuscript revision and additional experiments are necessary.

We thank the reviewer for the overall positive view of our findings that altered mPFC electrophysiology is associated with poor mPFC-dependent decision-making in the 3xTgAD model. We agree that the strength of our conclusions is somewhat limited by the scope of the experiments undertaken. We also agree that further efforts to reveal the mechanisms behind our observations are important. However, these would represent a set of major studies, particularly as our understanding of the impact that AD-like pathology has on mPFC is at a very early stage when compared to regions such as the medial

temporal lobe and, in particular, the hippocampus. Thus, we feel that the present paper is a worthy first step that will motivate interest in this research area and that our present correlative work stands on its own. Nevertheless, we have added a short section to the Discussion (lines 307 to 315) to highlight how these further studies might evolve, particularly, as stated by this reviewer, the requirement to show altered physiology and behaviour in the same animals.

Main point

The authors' statement is that, differently from control mice, AD mice fail to implement an advantageous strategy. However, although both P1 and P2 are advantageous, only P2 is preferentially chosen by control mice. Thus, as far as P1 is concerned, AD mice behave as do control mice, which means that they do not have a global executive dysfunction. It is therefore mandatory to understand why P2 is preferentially chosen by control mice compared to P1, and why AD mice do not do the same, knowing that the main difference between P1 and P2 is not in the number of rewarded or punished responses, but in the quantity of milk delivered. Thus, even though the authors claim that motivation is unaltered in AD mice, being the quantity of milk delivered apparently irrelevant for AD mice, some difference between genotypes in reward systems cannot be excluded. Accordingly, the interpretation of behavioral differences between genotypes needs to be re-oriented considering the respective motivational/rewarding values of P1 vs P2, and mice have to be tested in another task assessing executive functions with a negative reinforcement (or suppression of reinforcement, e.g. extinction) to estimate in which conditions their executive functions are impaired.

We can see the reviewer's point that control and AD mice both choose P1 to a similar extent. However, this ignores the fact that control mice choose P2 much more frequently and P1, P3, and P4 less frequently than P2. This is a clear sign that control mice can perform the task well, as P2 is the most advantageous possible choice (i.e., even more advantageous than P1). The choices of AD mice, on the other hand, are similar for P1, P2, P3 and P4, thus, they exhibit no clear preference and, as such, perform the task very poorly. We maintain, therefore, that simply comparing P1 choice percentage between control and 3xTg mice and concluding that 3xTg have no executive deficit would be misleading as it would ignore a substantial portion of relevant data. In addition, although the reviewer is correct in stating that the quantity of milk delivered differs between P1 and P2 choices, the number of rewarded and punished responses also differ.

We do not discount that some changes to reward/motivation may contribute to behavioural differences; however, our observed similarities between control and 3xTg mice in the number of omissions, premature responses, and the time taken for mice to reach training criterion strongly support an executive deficit as opposed to altered reward/motivation/reinforcement learning. We have added an extra sentence to the start of the Results section (lines 95-96) to further emphasize this.

Minor points

The 4CGT is a complex task which apparently engages mice for months. Despite the description of the experimental procedure in both the Results and Methods sections, it is unclear what happens between 3 and 6 months (Mice began staged training at 2-3 months of age and were fully trained to criterion by 6-7 months of age (n= 5 controls, n= 7 3xTgs). Does this mean that the training phase goes on from 3 to 6/7 months ? That there is some pause during the training stage

? In other words, how many weeks training consists in, and at what age weekly sessions are administered ?

The training phase for this task is long, containing multiple stages, and mice took 2-3 months of daily sessions to reach the pre-test criterion (to be precise, between 23 and 63 sessions, 5 days a week). There were no lengthy pauses in the training schedule. Mice, therefore, undertook training stages between the ages of 3 months and 6 months, and then began testing at 6 months, which required around 1 month (hence, mice were 6/7 months of age at the point of testing). In our experience and from reports in the literature, touchscreen tasks have long training stages. Prior to this study we conducted a pilot using C57 female mice at a similar age. These C57s were slightly quicker to learn the task than 3xTg control mice on a mixed c57/129sv background, but C57s still took between 17 and 44 sessions (i.e., 1-2 months of training). We have provided our comparison of C57 training with 129sv control mouse training (please see below) to show that, in the case of both control strains, training stages for the 4CGT took a couple of months to complete. We believe that the 129sv background on which the 3xTg strain is based may contribute to slightly longer training duration than C57 mice as this background has previously been linked with slower cognitive learning and poorer behavioural performance (<https://link.springer.com/article/10.1007/s002130050327>).

Histograms and curves summing up advantageous (P1+P2) vs disadvantageous (P3+P4) choices have to be removed from Figure 2 given that P1 choices do not vary between genotypes so that these data “hide” the fact that AD mice show regular advantageous P1 choices.

These data do not hide anything, their purpose is simply to show that control mice (but not 3xTg) choose the two most advantageous options significantly more than the two most disadvantageous options. This comparison is undertaken frequently in the literature for this task , for example

<https://www.sciencedirect.com/science/article/pii/S0028393208000651?via%3Dihub#aep-section-id28>.

We also include the breakdown of choice percentages in the same figure.

Electrophysiology

The authors should speculate more on why the deficit in input/output responses is layer II/III specific while the deficit in pair-pulse response is layer V- specific. More references to the anatomy of ventral hippocampal projections to, and anatomical properties of layer II/III vs Layer V in, infra-limbic (IL) cortex are needed. Also the dysregulation of E/I balance needs to be supported by data showing how interneurons contribute. For example, the statement that “Observed layer-specific differences in the response amplitude may, therefore, be associated with differences in local interneuron circuitry” absolutely requires to perform vasoactive intestinal polypeptide (VIP) and parvalbumin (PV), parvalbumin immunohistochemistry in IL slices.

We agree with the reviewer that more information regarding the layer circuitry may be useful to explain layer specific effects, however, we do include information on layer-specific circuitry in the discussion (lines 248 to 256) and do not want the discussion to be any longer than necessary (its length has already been flagged by the reviewers). We do have a figure that may help explain the microcircuitry (please see below) and would be happy to add this to the manuscript if the reviewers agree. Furthermore, we also agree that specific studies in neuronal subtypes, for example using patch-clamp electrophysiology or staining, is an excellent next step to elucidate potential mechanisms for our observations

and have included this suggestion at the end of the discussion (lines 307 to 315).

Gene expression alterations

This experimental part is carefully executed but largely non-specific to the manuscript topic for two reasons. First, changes in gene expression are measured in global mPFC extracts and hence do not inform on how these genes are dysregulated in the infra-limbic cortex. Second, the detected gene alterations are not specific to AD (see Tripathi, M.K et al. Sci Rep 14, 10 (2024). <https://doi.org/10.1038/s41598-023-50248-4>. [doi.org]) In my opinion these data could be suppressed from the present manuscript to be the object of an independent publication.

We thank the reviewer for this insight and agree that the RNA seq data, although promising, are somewhat limited in scope at the present time. As suggested, we have removed this section from the paper.

General comments

All sections of the manuscript are too long and need to be shortened.

The discussion largely consists in simply commenting the findings without addressing their main interest (does the 4CGT task reveal reward system alterations or executive function deficits in AD mice ?) or their translational potential to diagnose or cure AD.

We have tried to make the manuscript more precise and condense the Discussion in particular. Removing RNA sequencing data has also allowed us to shorten significantly all sections of the manuscript. We have added a comment on the issue raised re. altered reinforcement ('reward') vs executive function to the Discussion (lines 299 to 300), and potential future directions that will allow a more specific insight into how this circuitry is altered in AD (lines 307 to 312). We have also included a few lines into how this study may help to inform early AD diagnosis and treatment (lines 312 to 315).

Response to Reviewers

We would firstly like to thank both reviewers for their work in re-evaluating and improving the manuscript further. We have addressed their comments below. We have referred to any changes to the manuscript by their line numbers, and have also highlighted these changes in yellow in the manuscript file, to make them easier to identify.

Reviewer #1 (Remarks to the Author):

Point 1. Why did twice as many ‘control’ subjects fail to reach criteria than transgenic mice? This is worrisome and is not reasonably explained in the manuscript.

We understand the reviewer’s concerns regarding the number of mice that failed to reach training criterion. In general, this was due to a variable number of trial completions across sessions (i.e., mice were not completing enough trials per session), rather than error rates (task difficulty).

To more clearly show this, we have provided a comparison of 3xTg 129sv control vs C57 pilot data in training and testing stages (please see the figure below). From this, it can be concluded that:

- 1) 129sv mice appeared less motivated to perform the task; they had a significantly higher omission rate and completed significantly fewer trials per session than C57 mice.**
- 2) Despite this, 129sv mice did not require significantly more total sessions than C57 mice to train to criterion on the task.**
- 3) 129sv mice performed similarly to C57 mice during testing stages. In fact, 129sv mice were able to better distinguish between the two most advantageous options (P1 vs P2) better than the C57s, with P2 being the most advantageous option. They therefore performed slightly more efficiently than C57s.**

We have added this figure to the methods section of the manuscript (supplementary figure 2, page 12). We hope that the reviewer feels this is appropriate and adds clarity to the study. We also believe this may be useful in general for the field by releasing further strain data for mouse performance on the 4CGT, as there currently isn’t a high quantity of papers testing this. We have also added some information on page 12 to clarify that mice were excluded from testing due to low trial completion rate rather than poor learning.

Supplementary figure 2: Performance of C57BL/6 (C57) and C57/129sv (129sv) control mice on the 4CGT. (A) Comparison of the number of sessions taken to reach criterion at each training stage, and overall, between C57 and 129sv mice **(B-D)** Comparison of the **(B)** % omissions, **(C)** % premature responses, and **(D)** number of trials per session, between C57 and 129sv mice. **(E)** Average choice % of C57 mice by session. **(F)** The same as for **E**, but with sessions displayed in blocks of 4. **(G)** Average choice % of C57 mice for advantageous (P1+P2) vs disadvantageous (P3+P4) choices, by session block. **(H)** Average choice % of 129sv mice by session. **(I)** The same as for **H**, but with sessions displayed in blocks of 4. **(J)** Average choice % of 129sv mice for advantageous (P1+P2) vs disadvantageous (P3+P4) options, by session block. **(K)** Net advantageous choice % by individual session. This was calculated by subtracting the average % of disadvantageous choices (P3+P4) from the average % of advantageous choices (P1+P2) selected by mice across the session. The more positive the value, the better the decision-making and task performance. **(L)** The same as for **K**, but with sessions blocked. Both strains were able to implement an advantageous strategy and choose a higher % of advantageous than disadvantageous choices. n= 10 female C57 mice, n= 5 male 129sv mice. Error bars indicate SEM. *p<0.05, **p<0.01, ***p<0.001, ****p<0.0001.

Point 2. The Reviewer strongly disagrees that differences between conditions in the end result obviate power calculations. This is not rigorous. Most behavioral tests require much larger cohorts, especially with the difficulty in learning the task increasing, not decreasing the number of cases required to validate behavior. At a minimum, the number of dropped cases needs to be validated by independent means. A second behavioral test for executive function could be utilized but at least one well-powered method is necessary to draw conclusions for degeneration of executive functional behavior prior to overt pathology.

We feel that this point generally concerns our decision process when selecting initial sample sizes (which can often be informed by a power analysis). The extreme lack of data for 4CGT mice in general meant an *a priori* power calculation was not possible from published data. To be confident that mice could learn and perform the task under test conditions in our hands we ran a 4CGT pilot using 10 female C57 mice (see new Suppl Figure 2). We chose n=10 as this sample size is quite common in behavioural studies. Results showed that n=10 was a suitable cohort size to see good training and testing performance in neurotypical c57 mice, with no mice having to be excluded. We then based our experimental study on this cohort size, reduced to n=9 per group due to the limited availability of our age-matched transgenic 3xTgAD mice. Due to strain differences in motivation (see Point 1), we had to exclude some of our mice from both groups during training when applying the same exclusion criteria as in the pilot. Note that, for consistency, these criteria match those used across labs and touchscreen tasks in mice (and are as recommended by Campden Instruments/Lafayette). We have added a summary of the pilot study to support the required group sizes argument to page 11, lines 330-332. Unfortunately, we no longer hold the 3xTg strain in Manchester and so cannot perform further or repeat testing.

A power analysis (G*Power 3.1.9.6 for Mac) of our data to estimate required sample sizes *post hoc* reveals the following:

Controls (mean and SD data from Fig 2c session 2)

Calculated Cohen's d = 2.0

Using alpha=0.05, power 0.8;

CONTROLS: 2-tailed t-test would require n=5 (actual power = 0.91)

3xTgAD (mean and SD data from Fig 2f session 2)

Calculated Cohen's d = 0.35

Using alpha=0.05, power 0.8;

3xTgAD: 2-tailed t-test would require n=55 (actual power = 0.80)

These calculated values support our relatively small sample size as effective for controls at our required power and that a VERY large number of 3xTgAD mice would be required to see a similar difference, consistent with our large observed group difference in

performance. Thus, we feel that these analyses justify the final sample sizes in our behavioural analyses.

We do appreciate and agree that small group sizes for behaviour after exclusion criteria are applied are not ideal in general and have added a statement to the Discussion (lines 309-313) supporting further replicative studies in this and other AD models for the 4CGT. The fact that threshold effect size was still achieved in our study, despite a reduction in sample sizes at test after training exclusions, may be explained by the relatively stronger performance of our 129sv controls compared to c57 females (stronger preference for the most advantageous choice (P2) in 129sv; compare Suppl Fig 2f vs 2i).

Point 3. The reviewer agrees that the RNAseq data could be eliminated from the manuscript.

We thank the reviewer for confirming this.

Reviewer #2 (Remarks to the Author):

The ms has been revised by taking the reviewers points adequately. This new version is about ready for publication. Just two minors points remain that can be easily addressed.

We thank the reviewer for their positive view that the manuscript is almost ready for publication.

Point 1. It is necessary to include a small sentence in the discussion to explain why, among the two advantageous choices, control mice choose considerably more P2 than P1, making it the advantageous P1 chosen with the same frequency as the two disadvantageous P3 and P4. Maybe the difference in the amount of milk delivered (the double in P2 than in P) is the crucial factor which makes P2 overshadowing P1. Anyway, this huge difference between the two advantageous choices emerges so clearly in Fig2b that it is impossible not to comment it.

We agree with the reviewer that we should add a comment referencing the difference in control choice % between the two advantageous options, P1 and P2. We believe this is most likely due to the ability of the mice to recognise that P2 dispenses more milkshake over the course of the session. We have added this detail in lines 181-184 of the discussion.

Point 2. The discussion includes sub-headings for electrophysiological data which undoubtedly facilitate the reading. I suggest to follow the same formula all along the discussion with a sub-heading "Executive functions" at line 174, and "Conclusions" at line 297.

We agree that further subheadings in the discussion will help to improve readability and thank the reviewer for this suggestion. We have added the suggested subheadings of "Executive functions" and "Conclusions" on lines 174 and 300, respectively.